# The impact of resection margin distance on survival and recurrence in pancreatic ductal adenocarcinoma in a retrospective cohort analysis

**Dennis Obonyo[1], Verena Nicole Uslar[1]\*, Johanna Münding[2], Dirk Weyhe[1], Andrea Tannapfel[2]**

1 University Hospital for Visceral Surgery, Carl von Ossietzky University Oldenburg, Oldenburg, Germany,
2 Institute for Pathology, Ruhr University Bochum, Bochum, Germany

\* verena.uslar@uol.de

## Abstract

### Background

The prognostic effect of resection margin status following pancreatoduodenectomy for pancreatic ductal adenocarcinoma (PDAC) remains controversial, even with the implementation of standardized pathological assessment. We therefore investigated the impact of resection margin (RM) status and RM distance in curative resected PDAC on overall survival (OS), disease-free survival (DFS) and recurrence.

### Method

108 patients were retrieved from a prospectively maintained database of a certified pancreatic cancer center. Distribution and relationships between circumferential resection margin (CRM) involvement (CRM≤1mm; CRM>1mm; CRM≥2mm) and their prognostic impact on OS and DFS were assessed using Kaplan-Meier statistics and the Log-Rank test. Multivariate logistic regression was used explain the development of a recurrence 12 months after surgery.

### Results

63 out of 108 patients had medial RM and 32 posterior RM involvement. There was no significant difference in OS and DFS between CRM≤1mm and CRM>1mm resections. Clearance at the medial margin of ≥2mm had an impact on OS and DFS, (RM≥2mm vs. RM<2mm: median OS 29.8 vs 16.8 months, median DFS 19.6 vs. 10.3 months). Multivariate analysis demonstrated that age, medial RM ≥2mm, lymph node status and chemotherapy were prognostic factors for OS and DFS. Posterior RM had no influence on OS or DFS.

### Conclusion

Not all RM seem to have the same impact on OS and DFS, and a clearance of 1mm for definition of a negative RM (i.e. CRM>1mm) seems not sufficient. Future studies should include more patients to stratify for potential confounders we could not account for.

**Data Availability Statement:** Data cannot be shared publicly because of the potential risk of deanonymization. Data are available from the

institutional Ethics Committee (contact via Email: med.ethikkommission@uol.de or telephone: +49 (0) 441 798-3109) for researchers who meet the criteria for access to confidential data.

**Funding:** The author(s) received no specific funding for this work.

**Competing interests:** The authors have declared that no competing interests exist.

**Abbreviations:** CI, Confidence Interval; CRM, Circumferential Resection Margin; CTx, adjuvant Chemotherapy; IPMN, Intraductal Papillary Mucinous Neoplasms; LN, Lymph Node; PD, Pancreaticoduodenectomy; PDAC, Pancreatic ductal adenocarcinoma; PPPD, Pylorus-Preserving Pancreatoduodenectomy; PV, Portal Vein; RCP, Royal College of Pathologist; RM, Resection Margin; SMA, Superior Mesentery Artery; SMV, Superior Mesenteric Vein.

## Trial registration

This study was registered with the German Clinical Trials Registry (reference number DRKS0017425).

## Introduction

Pancreatic ductal adenocarcinoma (PDAC) is the seventh leading cause of cancer-related deaths worldwide, and in Europe the fourth most deadly cancer after lung, colorectum and breast cancer [1, 2]. So far surgery with complete tumor-free resection margins (R0) combined with adjuvant chemotherapy offers the only chance for a cure, though with a dismal 5-year survival ranging from 5–17% [3, 4].

Tumor free resection margins generally serve as a long-term prognostic factor [5, 6]. A clear definition of resection margin status is therefore essential [7]. According to many centers in Europe, the R classification was adapted to Royal College of Pathologist (RCP) guidelines [8]. According to the German clinical practical guidelines R0-resected PDAC are called CRM positive (Circumferential Resection Margin) if there are tumor cells within 1 mm of the RM (CRM≤1mm), but not at the margin (R0 narrow) and CRM negative (CRM>1mm, R0 wide) refers to RM with no tumor cells within 1 mm of the definitive RM. Whereas R1 (<0mm) implies when tumor cells are found at the definitive RM [9]. There is accumulating knowledge that clear resection margins (R0 > 1mm) has a significant impact on survival following PDAC resection [5, 10–12]. Nevertheless, exact definition of a margin-free resection (R0) and its prognostic significance on survival in resected PDAC remains internationally controversial. Thus, the rates of reported R1 resection differ in literature between 13% and 85% [11, 13–16], and vice versa [17]. Numerous studies have reported that resection margin or CRM involvement is an independent prognostic factor for poor long-term survival [5–7, 18, 19]. Dispute exists as other several studies have failed to demonstrate a survival benefit for patients with a margin-negative resection [20–22]. Furthermore, almost half of the resected patients, irrespective of adjuvant treatment administered, experience local or systemic disease relapse only 12 months after surgery. This implies that most patients had a systemic disease even in those few cases of reported R0 resections [23–25]. Therefore, a resection margin distance of 1 mm might not be enough to promote long-term survival in PDAC.

The aim of this study was to investigate the impact of different resection margins in PDAC on OS and DFS via Kaplan-Meier statistics, and on disease recurrence 12 months after surgery by implementing a multivariate logistic regression model, using the CRM distances ≤ 1 mm, > 1 mm, and ≥ 2mm. Our hypotheses where that not all resection margins equally impact on survival, and that a resection margin distance as defined in the German guideline (CRM positive vs. CRM negative) might not be enough to explain long term survival, but rather that larger distances should be achieved.

## Materials and methods

### Study design

The study was approved by the medical Committee for Research Ethics at the University of Oldenburg (reference number: 2019–071) without the need for Informed Consent due to the retrospective nature of this study, and was registered with the German Clinical Trials Registry (reference number DRKS0017425). It was in compliance with the Helsinki Declaration.

For this retrospective study, 478 consecutive patients of the certified pancreatic cancer center of the Clinic of General and Visceral Surgery, Pius Hospital Oldenburg, University of Oldenburg with a histological diagnosis of PDAC who were scheduled for curative pancreatic resection between 01/2010 and 12/2019 were screened for inclusion from an electronic institutional prospectively maintained database (see also Fig 1). Data were collected until 01/2021 to allow for at least one year of follow-up. Operative data and patient characteristics were collected in the database, including age at time of surgery, sex and type of surgery. Tumor characteristics recorded were: pTNM stage, grade, histopathological diagnosis, lymph node (LN) involvement, total number of examined lymph nodes (ELN), lymph node ratio (LNR), and lymphangioinvasion and/or perineural invasion. All the periampullary carcinomas, bile duct carcinomas, body/tail carcinomas, adenocarcinomas arising in the presence of intraductal papillary mucinous neoplasms (IPMN), neuroendocrine carcinomas and cancers of uncertain origin were excluded. All R2 resections were excluded. Also excepted from the study were patients who died within 30 days after resection (30-day mortality). Patients who underwent neoadjuvant chemotherapy were excluded in this study. Thus, out of the 478 initially screened patients 108 patients with resected PDAC of the pancreatic head were included in the analysis.

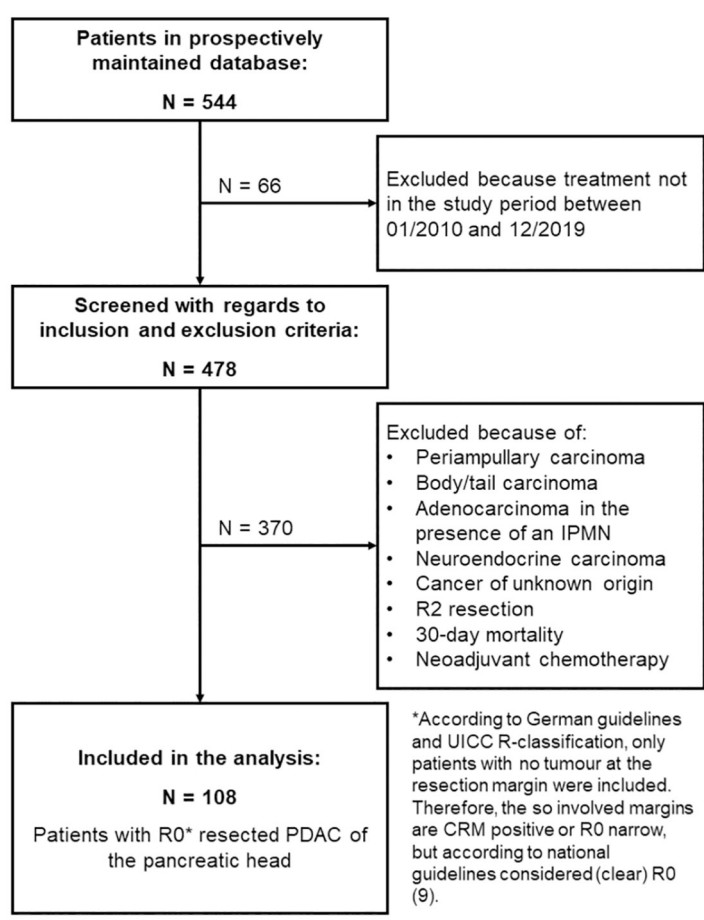

**Fig 1. Flow chart depicting the screening process for inclusion of patients.**

## Surgical procedure and postoperative treatment

After standard preoperative evaluation, decision to offer surgery was made by the multidisciplinary team of the tumor board. Both computer tomography (CT) or magnetic resonance tomography (MRT) were used during discussion in the tumor board. Very rarely was endoscopic ultrasonography used to formulate the final decision. The criteria for respectability were; I) absence of distant disease, II) absence of locally advanced disease, e.g. involvement of superior mesentery artery (SMA), common hepatic artery or coeliac trunk axis or involvement of the portal vein (PV)/ superior mesenteric vein (SMV) up to 180˚. For patients with intraoperative local advanced disease adjacent to either SMV or PV greater 180˚, resection and reconstruction of the SMV/ PV, or superior mesenteric-portal vein confluence were performed if a R0 resection was feasible (n = 4). The pancreatic neck margin was routinely submitted for intraoperative histological examination, and if suspicious tumor cells were found, a re-resection or a total pancreatectomy (n = 5) was performed. Out of those, n = 4 were positive at the medial and n = 1 was positive at the posterior margin.

Our pancreaticoduodenectomy (PD) surgical procedure has been described earlier and has not changed [7]. The pylorus-preserving pancreatoduodenectomy (PPPD) was the first choice of procedure and not the classical Kausch-Whipple procedure. After surgical resection, every patient was again discussed by the multidisciplinary tumor board to determine whether adjuvant therapy was indicated. Adjuvant chemotherapy (CTx) was recommended for all patients after curative resection (R0 and R1). Chemotherapy with gemcitabine and capecitabine or gemcitabin monotherapy or modified 5-fluorouracil, folinic acid, irinotecan, and oxliplatin combination (mFOLFIRINOX) was the standard.

## Histopathogical assessment

The routine histopathological examination was performed for every patient according to German national guidelines and RCP guidelines for the purpose of comparability [8, 9]. This examination is a standardized procedure performed for all specimen resected in the Clinic of General and Visceral Surgery of the Pius-Hospital by the institute for Pathology in Bochum. As recommended by German national guidelines we analyzed the transection margins and the CRM. Thus, the following margins were examined in our study: the transection margins which includes pancreatic neck margin, bile duct margin, proximal duodenal/stomach margin and distal duodenal margin as well as proximal and distal vessel margins, (in case of vascular resection) as well as the CRM; which included the posterior margin and the medial margin (see also Fig 2). The medial margin was defined as the groove along the superior mesenteric vein/portal vein and the transection surface when dissecting the uncinate process from SMA, i.e. the surface that segments of the SMV, PV, or SMA are found. In our study, we did not consider the anterior surface as a true surgical margin, and excluded it from routine inking. In case of tumor infiltration of the anterior pancreatic surface inking was performed [9, 14]. All PD specimens were fixed in 10% buffered formalin. The circumferential soft tissue margins were stained according to recommendations of RCP. The specimens were then sliced in 3- to 5-mm-thick slices following an axial plane perpendicular to the duodenal axis. The tumor stage was determined using the current UICC TNM classification system, 8th edition, [26].

## Data acquisition

All clinicopathological data including histology, LN status, and tumor type and tumor stage are obtained prospectively by specialized tumor documenters from the clinical and pathological records. Patients whose death was clearly documented as attributable to pancreatic cancer

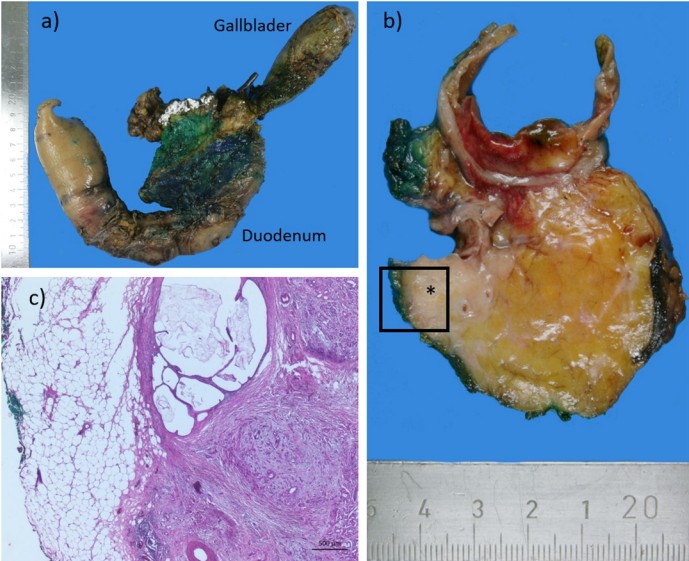

**Fig 2. A pancreaticoduodenectomy specimen that was inked and assessed using the axial slicing technique.** a) Color-marked Whipple resection (from dorsal); blue: retropancreatic area; green: vascular groove; white: pancreatic section margin. b) Cross-section of Whipple resectate with color-marked resection margins with the asterisk marking the pancreatic carcinoma in macroscopic section preparation. c) histological section preparation, distance to resection margin 0.2 cm (R0 wide, CRM negative).

were considered to have died of that disease; other deaths were not considered to have been caused by PDAC. Clinical follow-up data were obtained by reviewing the hospital records and by direct communication with the attending physicians in a standardized and structured manner based on the data sheet for pancreatic cancer centers of the German Cancer Society by the responsible tumor documenters. Overall survival (OS) was calculated from the date of surgical resection of the tumor to the date of death or last follow-up (censored). Additionally, disease-free survival (DFS) was calculated, with data for patients who died from other causes being censored at their time of death or last follow-up.

Follow-up information was obtained from an electronic institutional database, the primary care physician or the referral oncologist. There is no scientific evidence that structured follow-up in terms of performing regular staging examinations leads to improved survival in pancreatic cancer [9]. However, in our clinic, clinical and tumor-maker check-ups are performed regularly. In case of clinical suspicion or elevated tumor-makers an imaging is then performed. In the case of elevated preoperative CA19.9 levels the assessment of this marker is performed every 3 months for 2 years and an abdominal CT scan every 6 months—according to ESMO Practice Guidelines for diagnosis, treatment and follow-up of Pancreatic adenocarcinoma [27]. Follow-up imaging (primarily CT or MRT) was performed when indicated clinically; no standard protocol was used. Recurrence was diagnosed based on evidence of disease recurrence on CT or MRT imaging. Local recurrence (LR) was defined as the presence of disease in the surgical bed or present in the mesentery, adjacent retroperitoneal soft tissue or around the SMA, common hepatic artery or coeliac trunk axis or around the PV/ SMV. Distant recurrence was defined as the presence of disease e.g. in the liver, peritoneum, omentum, lungs or aorto-intercaval lymph nodes. Early recurrence was defined as recurrence occurring within 6 months after surgery [28, 29]. In the event of recurrent disease, the opinion of the tumor board was sought regarding palliative therapy.

## Statistical analysis

IBM SPSS Statistics 26 for Windows was used for statistical analysis. Patient characteristics were analyzed descriptively, with means and the range or 95% confidence interval, or numbers and proportions given as appropriate.

OS and DFS was analyzed using the Kaplan-Meier method, and the 95% Confidence Intervals (CI) were used to estimate the effect. Since sample size depended on the number of patients available in the database, no a priori power calculation was performed. For the purpose of the analyses, patients were either grouped by involvement of the respective margin (yes; no), distance between tumor and resection margin (<1 mm; 1 to 2 mm; ≥ 2 mm), and number of resection margins involved (no margin involved; 1 margin involved; ≥ 2 margins involved).

To accommodate for the exploratory nature of this study, and to avoid multiple testing, no p-values were calculated. Thus, only CIs are given here. This is in accordance with various recommendations stressing the importance of reporting CIs in the context of clinical relevance [30–32]. In this study, we define a difference in survival as clinically relevant if there is a median difference of more than 12 months and as statistically significant if, in addition, the 95% confidence intervals do not overlap. Twelve months were chosen because we argue that such an increase in survival should indeed be viewed as relevant, especially in pancreatic cancer, and it also reflects appropriate caution with regards to the sample size of this study.

To analyze the influence of various variables on the development of a recurrence after 12 months, a multiple logistic regression analysis with stepwise backwards exclusion was performed for 80 out of the 108 patients. Nine input variables were included (Age at time of surgery (years), sex (male/female), pT, pN, pM, LKR, distance to the posterior resection margin (<1 mm; 1 to 2 mm; ≥ 2 mm), distance to the medial resection margin (<1 mm; 1 to 2 mm; ≥ 2 mm), and administration of systemic therapy (yes/no)), which roughly matches the rule of thumb that no more than one variable per 10 cases should be included in a logistic regression analysis [33].

## Results

As seen in the flow-chart (Fig 1) 108 patients underwent curative surgical resection because of PDAC of the pancreatic head. Only one patient underwent classical PD and 107 patients PPPD. 30-day mortality for the whole patient collective of the data base was 10.6% (n = 58 out of 544). These patients were excluded from analysis. Fifty-five patients were male (50.5%) and 54 female (49.5%). The mean age was 68.4 years (range 39 to 89 years) at the time of operation (see also Table 1). The median follow-up time was 17.2 months (range: 1–92 months). As reported in the table, if a minimum clearance of more than 1 mm (RM>1mm) is used, then about 28% of all patients fall in the R0 category. However, if historical UICC classification and national guidelines are applied (i.e., R0 > 0 mm), then 75% of all patients would be classified having a R0 resection.

### OS and DFS irrespective of positive margin site

There was no statistical difference as defined above in OS between patients with CRM>1mm (median OS 29.8 months, 95% CI: 20.9–38.8 months) and CRM≤1mm (median OS 18.6 months, 95% CI: 13.8–23.4 months).

A subgroup analysis of overall margin clearance by extending tumor free resection margin to 2 mm or more (i.e., < 1mm, ≥ 1 mm and < 2mm or ≥ 2 mm) showed that patients with more than 2 mm resection margin clearance (n = 76) had clinically relevant longer OS by about 15 months as compared to both groups with smaller clearance (n = 10, and n = 22, respectively; see Table 2).

**Table 1. Patient characteristics: Given are the mean and 95%CI, or the number and percentage, respectively.**

|  | All patients | SMV positive | SMV negative |
|---|---|---|---|
|  | N = 108 | N = 63 | N = 45 |
| Age | 68.4 (66.6; 70.2) | 68.2 (65.9; 70.5) | 68.8 (65.8; 71.8) |
| Sex (m/f) | 54/54 (50.0%/50.0%) | 27/36 (42.9%/57.1%) | 27/18 (60.0%/40%) |
| ASA |  |  |  |
| 2 | 28 (25.9%) | 28 (33.3%) | 7 (15.6%) |
| 3 | 79 (73.1%) | 79 (65.1%) | 38 (84.4%) |
| 4 | 1 (0.9%) | 1 (1.6%) | 0 (0.0%) |
| UICC |  |  |  |
| IA | 5 (4.6%) | 2 (3.2%) | 3 (6.7%) |
| IB | 9 (8.3%) | 3 (4.8%) | 6 (13.3%) |
| IIA | 12 (11.1%) | 5 (7.9%) | 7 (15.6%) |
| IIB | 63 (58.3%) | 42 (66.7%) | 21 (46.7%) |
| III | 14 (13.0%) | 8 (12.7%) | 6 (13.3%) |
| IV | 5 (4.6%) | 3 (4.8%) | 2 (4.4%) |
| pT |  |  |  |
| 1 | 7 (6.4%) | 2 (3.2%) | 5 (11.1%) |
| 2 | 22 (20.4%) | 8 (12.7%) | 14 (31.1%) |
| 3 | 78 (72.2%) | 52 (82.5%) | 26 (57.8%) |
| 4 | 1 (0.9%) | 1 (1.6%) | 0 (0.0%) |
| pN |  |  |  |
| 0 | 25 (23.1%) | 9 (14.3%) | (35.6%) |
| 1 | 68 (63.0%) | 45 (71.4%) | (51.1%) |
| 2 | 15 (13.9%) | 9 (14.3%) | (13.3%) |
| pM |  |  |  |
| 0 | 103 (95.4%) | 60 (95.6%) | 43 (95.2%) |
| 1 | 5 (4.6%) | 3 (4.4%) | 2 (4.8%) |
| LNR | 0.18 (0.15; 0.22) | 0.23 (0.18; 0.29) | 0.11 (0.082; 0.17) |
| R status |  |  |  |
| R0 (RM > 1 mm) | 30 (27.5%) | — | — |
| R0 (RM > 0mm) | 81 (75%) |  |  |
| 1 margin positive | 59 (54.1%) | — | — |
| ≥ 2 margins positive | 20 (18.5%) | — | — |
| CTX |  |  |  |
| not carried out | 29 (26.9%) | 16 (25.4%) | 11 (24.4%) |
| carried out | 74 (68.5%) | 44 (69.8%) | 29 (64.4%) |
| unknown | 5 (4.6%) | 3 (4.8%) | 5 (11.1%) |

pM1: Three solitary intraoperatively detected liver metastasis were resected and two patients had transverse mesocolon resection. These minor liver resections and mesocolon resection had no effect on mortality and survival rate compared to standard pancreaticoduodenectomy.

—: not reasonable to calculate

## Impact of posterior resection margin involvement

There was no significant difference in OS and DFS with regards to posterior resection margin involvement; median OS 20.4 months (95% CI: 13.3–27.5 months); and median DFS 12.6 months (95% CI: 10.8–14.5 months). Extending the margin clearance of the posterior margin

**Table 2. Overall survival (OS) and disease-free survival (DFS) irrespective of positive margin site as a function of margin distance.**

| | | Median | 95% CI | |
|---|---|---|---|---|
| | | | lower boundary | upper boundary |
| OS | <1mm | 17.2 | 12.5 | 22.0 |
| | ≥1mm and < 2mm | 15.9 | 6.0 | 25.9 |
| | ≥2mm | 31.0 | 19.4 | 42.6 |
| DFS | <1mm | 12.0 | 8.6 | 15.5 |
| | ≥1mm and < 2mm | 9.9 | 6.0 | 13.8 |
| | ≥2mm | 19.6 | 8.6 | 30.5 |

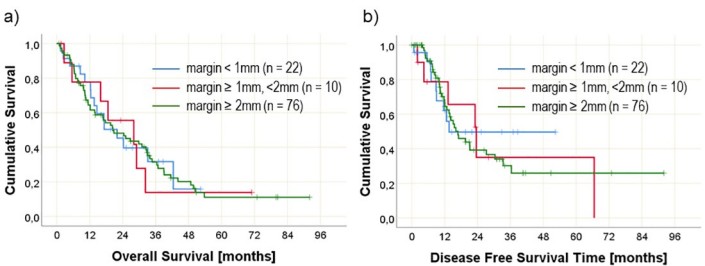

**Fig 3. Overall survival (left side) and disease-free survival (right side) over time in relation to involvement of the posterior resection margin for three different resection margin distances.** Numbers at risk are given in brackets.

**Table 3. Overall survival (OS) and disease-free survival (DFS) as a function of medial resection margin clearance.**

| | | Median | 95% CI | |
|---|---|---|---|---|
| | | | lower boundary | upper boundary |
| OS | <1mm | 17.1 | 10.9 | 23.4 |
| | ≥1mm and < 2mm | 13.6 | 7.4 | 19.9 |
| | ≥2mm | 29.8 | 20.6 | 39.1 |
| DFS | <1mm | 12.0 | 8.7 | 15.3 |
| | ≥1mm and < 2mm | 8.5 | 5.9 | 11.0 |
| | ≥2mm | 19.6 | 8.4 | 30.8 |

to 2 mm or more had no impact on OS or DFS. OS and DFS with respect to margin clearance are shown in Fig 3.

## Impact of medial resection margin involvement

Involvement of the medial resection margin reduces OS on a significant and clinically relevant level (median OS without involvement: 29.8 months, 95%CI 22.3–37.4 months; median OS with medial margin involvement: 16.9 months, 95% CI 13.0–20.8 months) and DFS differs on a clinically relevant level (median DFS without margin involvement: 30.4 months, 95% CI 11.1–49.7 months; median DFS with medial margin involvement: 14.2 months, 95% CI 12.0–16.5 months).

A resection margin clearance of more than 2 mm leads to a clinically relevant longer OS and remarkably longer DFS compared to patients with resection margin clearance less than 2 mm from the resection margin (Table 3 and Fig 4).

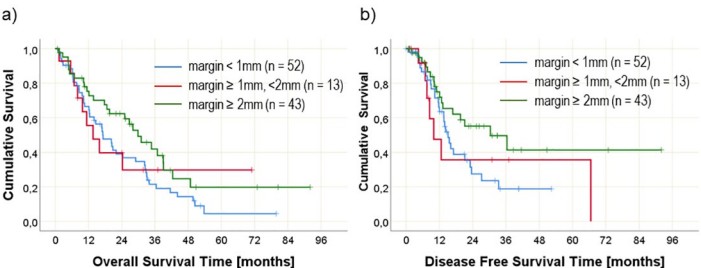

**Fig 4. Overall survival (left side) and disease free survival (right side) over time in relation to involvement of the medial resection margin for three different resection margin distances.** Numbers at risk are given in brackets.

## Impact of adjuvant chemotherapy and lymphnode status on survival

Adjuvant chemotherapy (CTx) was administered to 74 (68.9%) patients. CTx was not given in 27.5% patients due to poor performance status or patient refusal (n = 30). For 5 patients no information was available. OS was longer with CTx (median OS 28.0 months with vs 9.7 months without CTx) on a clinically relevant level.

There was also a clinically relevant difference in the median overall survival between N0 and N1 or N2 patients (34.7 months vs. 16.9 months vs 14.0 months).

## Involved margins

In the 108 patients, we detected no involvement of the bile duct transection margin and the jejunal resection margin. Pancreatic transection margin and gastric resection margin were each involved in 2 patients. Posterior resection margin involvement was found in 32 patients (29.6%), and medial resection margin involvement in 63 patients (58.3%). The anterior pancreatic surface was only in one patient microscopically involved. Further analyses therefore focused only on the posterior and medial resection margin.

30 patients (27.8%) had tumor free resection margins (CRM > 1mm). CRM positive resection was reported in 78 patients (72.2%), of which 57 patients had one positive margin (52.8%), 20 patients had 2 or more positive resection margins were found (18.5%), with the most common combination being a positive posterior and medial resection margin (n = 19; 17.6%), and one patient showed 3 positive resection margins (posterior, medial and gastric resection margin/ anterior pancreatic surface). For further analyses we grouped the patients as follows: no resection margin involvement, 1 resection margin involved, ≥ 2 resection margins involved.

When the posterior resection margin distance was analyzed, 22 patients showed a resection margin < 1 mm, 10 patients showed a distance between 1 and 2 mm, and 76 patients showed a distance ≥ 2mm. For the distance between tumor and medial resection margin, 52 patients had a resection margin < 1 mm, 13 had a resection margin between 1 and 2 mm, and 43 had a resection margin ≥ 2mm.

When the OS was analyzed as a function of the number of involved margins analyzed, the median OS for patients with no margin involvement, one, and two or more margins involved were 31 months (95% CI 19.4–42.6 months), 20.2 months (95% CI 12.7–27.8 months) and 15.9 months (95% CI 11–20.9 months), respectively. Thus, patients with margin positivity had a relatively shorter OS compared to patients with tumor free resection margins. There was no significant difference whether one or more margins are involved.

**Table 4. Amount of recurrences as a function of margin involvement\*.**

| | Count | Thereof with recurrence | Percentage of patients with recurrence |
|---|---|---|---|
| Without any margin involvement | 17 | 7 | 41% |
| Only posterior resection margin involvement | 7 | 1 | 14% |
| Only medial resection margin involvement | 19 | 11 | 58% |
| Posterior and medial resection margin involvement | 12 | 8 | 67% |

\*Only those patents with either posterior and/or medial resection margin involvement are regarded here

## Factors for the development of a recurrence

Out of the 108 patients, 80 patients had a follow-up period longer than 12 months. Out of those, 29 (36%) developed a recurrence at some point during these 12 months. Without margin involvement the recurrence rate was 41% (7 out of 17). If there was any margin involvement the recurrence rate was 35% (22 out of 63). However, there is a significant difference in recurrence rates, if recurrence rates are analyzed for each margin site separately (see Table 4). The Odds Ratio calculated for posterior vs medial margin involvement is 8.3 (95% CI 0.82–82.67), the Odds Ratio for posterior vs. posterior and medial margin involvement is 12.0 (95% CI 1.05–136.79).

Additionally, a multivariate logistic regression analysis was performed to develop a model explaining the development of a recurrence 12 months after surgery. All 80 patients were included in the analysis. The dependent variable was "development of a recurrence during the 12-month follow-up" (yes/no). Input variables included: Age at time of surgery (years), sex (male/female), pT, pN, pM, LKR, distance to the posterior resection margin (<1 mm; 1 to 2 mm; $\geq$ 2 mm), distance to the medial resection margin (<1 mm; 1 to 2 mm; $\geq$ 2 mm), and administration of systemic therapy (yes/no). A stepwise backwards approach yielded the best model with a specificity of 84.8% and a sensitivity of 62.1%, a Log-Likelihood of 65.7, and a Nagelkerkes $R^2$ of 0.499. This model included the variables age, pN, pM, medial margin distance und systemic therapy. None of the variables was significant be itself except age (see also Table 5).

**Table 5. Logistic regression model explaining the development of a recurrence during a 12 month follow-up.**

| | Regression coefficient | Standard error | p-value | Odds Ratio | 95% CI for Odds Ratio | |
|---|---|---|---|---|---|---|
| | | | | | lower boundary | upper boundary |
| Age at diagnosis | -0.07 | 0.04 | 0.040 | 0.929 | 0.87 | 1.00 |
| pN | | | 0.841 | | | |
| pN1 vs pN0 | | 8995.33 | 0.998 | 0 | 0.00 | . |
| pN2 vs pN0 | 0.55 | 0.94 | 0.557 | 1.733 | 0.28 | 10.85 |
| pM | -22.12 | 22517.33 | 0.999 | 0 | 0.00 | . |
| medial margin distance | | | 0.090 | | | |
| $\geq$1mm and < 2mm vs <1mm | -0.31 | 0.64 | 0.632 | 0.737 | 0.21 | 2.57 |
| $\geq$2mm vs <1mm | 2.42 | 1.29 | 0.061 | 11.265 | 0.89 | 142.15 |
| syst Therapy conducted | 1.54 | 0.83 | 0.063 | 4.669 | 0.92 | 23.64 |
| Constant | 26.17 | 22517.33 | 0.999 | 2.31055E+11 | | |

pN: pathological nodal stage

pM: pathological metathesis satge

## Discussion

In the last decade, there is a trend towards an international PD specimen reporting [34–37]. Nevertheless, the terminologies used to define RM or CRM and the heterogeneity of published R1 resections is debatable. The medial margin for instance is frequently defined as the mesenteric, SMV, SMA, vascular groove, uncinate, or even retroperitoneal margin [15, 38–41]. While the posterior margin is termed as uncinate, vascular groove or retroperitoneal margin or surface in other studies [42, 43]. Furthermore, the impact of different involved margins differs in literature [10, 24, 28, 44, 45]. Thus, the pathological reporting of involved margins in literature is inconsistent and confusing. As Haeberle et al. puts it, transection margins are clearly defined (pancreatic neck margin, bile duct margin, proximal duodenal/stomach margin and distal duodenal margin as well as proximal and distal vessel margins, in case of vascular resections), whereas international standardized nomenclature of different CRM lacks completely [46].

To reduce further confusion on nomenclature of CRM, we used in our study closely similar definitions described by Esposito et al and Jamieson et al [14, 45]. Thus, using the pathological specimen assessment according to modified RCP guidelines and German national guidelines, R1 as defined by the RCP resection rate was 75%. This confirms that most pancreatic resection are R1 [5, 11, 14]. The most frequently identified positive margin is the medial margin followed by posterior margin.

In our study, there was no significant difference in OS between the CRM≤1mm and CRM>1mm resections according to our used definition, though we could see a relevant clinical difference between the two groups of about 11 months, which in pancreatic cancer patients is not to be underestimated. However, patients with more than 2 mm resection margin clearance had an even more substantial, clinically relevant longer survival compared to R1 subgroup of more than 15 months, as compared to the difference in OS between the groups with <1mm margin and between 1mm– 2mm. For DFS we observe a similar effect, albeit not quite as large. Thus, the results of our study suggest that a margin clearance of 1 mm for definition of R1 might not be sufficient to predict OS and DFS as previously reported by Chang et al. and Gebauer et al. [15, 47]. The fact that we found no significant difference in survival between those patients with resection margins between 1–2 mm and those with R1≤1mm margins, suggests that these patients may have had occult tumor cells at the time of surgery. Tummers and colleagues had a questionable low rate of R1 (≤1mm) resection of 40.1% and found a significant difference in OS between the CRM negative and the CRM positive group in their cohort, 22 months vs. 12 months for CRM negative and CRM positive respectively [29]. In addition, their patient´s collective had a significant recurrence free survival, as they found DFS of 36 months for local recurrence or 20 months for distant recurrence compared to our study or Ghaneh et al. [5]. Nevertheless, they showed that the vascular resection margin is more often involved (52.3%) as compared to other margins, similar to our findings.

In our patients ´cohort, the medial margin positivity or involvement reduces OS on clinically relevant level (29.8 months vs. 16.9 months), whereas DFS was significantly prolonged in patients without margin involvement. Thus, medial margin seems to predict long-term survival for PDAC after PD as reported earlier by Zhang et al. [41]. Furthermore, almost 60% of patients with medial margin positivity had a recurrent disease during the follow-up period. In our study and similar to Zhang et al., the medial resection margin refers to the surface that SMV/PV and SMA are found. Therefore, our data support Pine et al. findings that SMV and SMA margin positivity are associated with poorer survival. Furthermore, a clearance of more than 2 mm at the medial resection margin leads to significantly longer DFS compared to patients with margin clearance less than 2 mm.

Similar to Pine et al., regarding the posterior margin our study shows no statistical significance on OS and DFS [10]. Interestingly, increasing the margin clearance of the posterior margin (i.e., > 2 mm) had also no impact on OS or DFS in our study. Delpero et al. showed reduced survival for patients with medial margin R1- resection, but not in the posterior margin involvement group [44]. Like Ghaneh et al. we showed that patients with RM clearance less than 1 mm of the transection margins (margins of bile duct, proximal gastric or duodenum, and jejunum) and pancreatic transection margin were not associated with significantly worse OS and DFS than patients with R0 > 1 mm or R0 < 2 mm tumor margins [5]. However, it has to be stated that there were too few patients with direct margin involvement (R1 direct/R1<0mm) to permit a separate analysis. In our institution, we routinely perform intraoperative pathological assessment of pancreatic transection margin and common bile duct, and if necessary, a re-resection is done. This could explain why there are a few patients with involvement of this margin in our study compared to other studies [5, 10].

The positivity of two or more margins has been reported previously to be 26–45% [10]. In our study, the rate of multiple margin involvement was 18.5%. We could see a clinically relevant poorer OS when multiple margins are involved, with OS decreasing by more than 15 months if 2 or more margins are involved as compared to no margin involvement, though not statistically significant. However, in cases of margin positivity, it does not matter much if 1 or 2 and more margins are involved, i.e. median OS 20.2 months vs. 15.9 months respectively.

As a known fact, resection margin status, (i.e., R0 or R1) neither has an influence on disease recurrence nor the pattern or site of disease recurrence also in our study [20, 29]. More than a third of the patients in the cohort followed up for at least 12 months after curative resection had a recurrent disease during this follow-up period. This result support other studies findings, in such that PDAC must be considered a systemic disease even in those few cases of apparent R0 resections [15, 20, 22–25].

Furthermore, when the different CRM regarding the impact on disease recurrence were assessed, patients with medial and not posterior CRM had a higher likelihood of having a recurrent disease. Incidences of bile duct transection margin, proximal duodenal and distal jejunal margin involvement were too few to make a meaningful analysis. Overall, medial margin involvement seems to be associated with the highest overall recurrence rate in our study with up to nearly 60%, whereas patients with only posterior resection margin involvement showed a remarkably low recurrence rate of < 20%. However, this might be at least in some part due to the low numbers in this group.

The effects of adjuvant therapy on overall survival have been studied extensively, but few studies have evaluated the association of adjuvant therapy with patterns of recurrence [23, 48]. In our study, patients who received chemotherapy had significantly longer survival compared with patients without chemotherapy (median OS 28.0 months with vs 9.7 months without CTx). In addition, CTx remained as an influencing factor in the logistic regression explaining recurrences after 12 months. We therefore expect CTx to reduce the likelihood of recurrence disease. This assumption may be supported by the updated 5 year survival analysis of the PRODIGE 24/CCTG PA6 Trial [49] However, the major limitations of this trial included the inability to capture the patients who were unable to recover adequately from the operation to receive adjuvant therapy and the inability to apply this effective but toxic regimen to all PDAC patients because the trial was highly selective. Because of the small number of patients in this study, we did not include patients treated with neoadjuvant therapy and therefore cannot conclude on the impact of neoadjuvant therapy or adjuvant CTx on recurrence rate or pattern. As reported earlier [7, 22], CTx seems to prolong OS in patients regardless of whether the resection margin is tumor-free or not.

A multifactorial logistic regression model with very good specificity and moderate sensitivity suggested that prognosis with regards to recurrence of PDAC depends on several factors concurrently. In this study, we found that medial margin positivity, chemotherapy and nodal status, along with age and pM status had the strongest influence on recurrence in a 12 month follow up. This is in line with our findings with regards to long-term survival [7]. These findings support the biologic heterogeneity of this tumor, suggesting multifactorial and simultaneous factors affecting long term survival and recurrence after resection of PDAC and not only RM status or lymph node status alone. Nevertheless, various studies have shown the impact of resection margin status on OS using the RCP definition of R1 $\leq$1mm [5, 7, 16–18], and in our study we can show, that the resection margin distance at the medial margin also has an influence on the recurrence. Since, although not significant, the recurrence rates are surely influenced on a clinically relevant level by the medial resection margin distance, as can be observed by the Odds Ratio for the recurrence rate between the group with <1mm margin clearance and with > 2mm margin clearance. Considering the diffuse growth pattern of PDAC and the high rates of recurrence, especially if the medial margin is involved, our data thus reflect the aggressive biological behavior of PDAC. Bearing this in mind, we don´t see any prognostic impact including R1 0mm (direct involvement) in the pathological assessment of PD specimens in future.

The major strengths of our study include the uniform assessment of PD specimens using the axial slicing technique as per RCP and German national guidelines for exclusively patients with PDAC of pancreatic head. Further, we analyzed the impact of different resection margins clearance on recurrence and OS in the setting of adjuvant chemotherapy. Our study has some limitations despite aforementioned strengths: Due to the retrospective single center nature of this study with a relatively small number of patients, the validity is somewhat limited. For example, reliable statements on the influence of the resection margins not discussed here are not possible. Also, although the effects of medial resection margin involvement seem quite clear, they have in some cases failed to reach significance. And the fact, that we did not find an effect of posterior resection margin involvement does not mean that there is no effect, due to the small sample size. Also, one could argue that the data used to calculate DFS are not very reliable. However, due to the procedure described above in the context of regular follow-ups, we are very likely to obtain reliable information on whether an actual recurrence occurred. However, the actual time of occurrence of the recurrence could be somewhat subject to error. Nevertheless, this would apply equally to all patients. Lastly, the margin distance is in some extent a surrogate of the tumour biology or rather the invasiveness of pancreas cancer in general, since larger and/or more invasive tumours are more difficult to resect with a wider resection margin. A case matched analysis would be needed to analyse this, but in our opinion this should be done on data gathered prospectively in a multicentric setting and in different pathologies. Therefore, a larger multicenter study is needed to confirm our findings on impact of different resection margins distance or clearance on recurrence and survival.

## Conclusion

Our study suggests that a medial resection margin of more than 2 mm is important for survival and prediction of recurrence following PDAC of pancreatic head. Thus, we argue that a margin clearance of 1 mm for definition of R1 seems not to be sufficient to predict OS and DFS, since the difference between the CRM$\leq$1mm and CRM>1mm resection was not clear. Considering the confusing terminologies used to define resection margins and the heterogeneity of published definitions for CRM, there is a need of international consensus on a standardized protocol for pathological assessment of PD specimens. Future studies should adopt uniform

reporting of pathological assessment and interpretation of margins in order to be able to compare results. Furthermore, future studies should identify patients with risk factors preoperatively, e.g. lymph node positivity and advanced tumors to the medial margin, in order to evaluate the benefit of neoadjuvant therapy and improve DFS.

## Acknowledgments

The authors wish to thank Fynn Piastowski, a student assistant, who helped with data curation.

## Author Contributions

**Conceptualization:** Dennis Obonyo, Dirk Weyhe, Andrea Tannapfel.

**Data curation:** Dennis Obonyo, Verena Nicole Uslar, Johanna Münding.

**Formal analysis:** Verena Nicole Uslar.

**Investigation:** Verena Nicole Uslar.

**Methodology:** Dennis Obonyo, Verena Nicole Uslar, Johanna Münding, Dirk Weyhe, Andrea Tannapfel.

**Project administration:** Dennis Obonyo, Dirk Weyhe, Andrea Tannapfel.

**Resources:** Dirk Weyhe, Andrea Tannapfel.

**Supervision:** Dirk Weyhe, Andrea Tannapfel.

**Validation:** Dennis Obonyo.

**Visualization:** Verena Nicole Uslar, Johanna Münding.

**Writing – original draft:** Dennis Obonyo, Verena Nicole Uslar.

**Writing – review & editing:** Dirk Weyhe, Andrea Tannapfel.

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
