## [Decision Letter · Decision Letter 0]

5 Dec 2022

PONE-D-22-07704The impact of resection margin distance on survival and recurrence in pancreatic ductal adenocarcinoma in a retrospective cohort analysisPLOS ONE

Dear Dr. Uslar

Thank you for submitting your manuscript to PLOS ONE. After careful consideration, we feel that it has merit but does not fully meet PLOS ONE’s publication criteria as it currently stands. Therefore, we invite you to submit a revised version of the manuscript that addresses the points raised during the review process.

We look forward to receiving your revised manuscript.

Kind regards,

Fabian Finkelmeier, M.D.

Academic Editor

PLOS ONE

Journal Requirements:

Reviewers' comments:

Reviewer's Responses to Questions

**Comments to the Author**

1. Is the manuscript technically sound, and do the data support the conclusions?

Reviewer #1: No

Reviewer #2: Yes

2. Has the statistical analysis been performed appropriately and rigorously? 

Reviewer #1: No

Reviewer #2: Yes

3. Have the authors made all data underlying the findings in their manuscript fully available?

Reviewer #1: Yes

Reviewer #2: Yes

4. Is the manuscript presented in an intelligible fashion and written in standard English?

Reviewer #1: Yes

Reviewer #2: Yes

5. Review Comments to the Author

Reviewer #1: 1. In the introduction, the authors report 5-year survival for resected pancreatic cancer followed by adjuvant therapy as 17%. Would suggest a range especially that they cited the PRODIGE trial where the 5-year survival for the FOLFIRINOX arm was higher than 17%

2. With more centers moving to neoadjuvant therapy for all stages of pancreatic cancer, it might be helpful to have this analysis done in this group of patients. That’s said, this paper is still valuable to patients with upfront resection.

3. 30 day mortality of 10% is a bit worrisome. Can the authors provide why it’s higher than expected.

4. R0 of 28% is low but not surprising in upfront resected patients. Another reason why neoadjuvant should be given and why it would be interesting to do this study in this group of patients.

5. Metastatic patients should not be included in this analysis. Margin status is irrelevant in patients who already have metastatic disease

6. I do understand why the authors do not want to use p value but comparing two confidence interval is another way of doing inferential statistics. In addition, choosing 12 months as clinically relevant cut off for survival analysis is arbitrary and not based on any scientific merit. Since when doubling survival in cancer is the cut off for clinically relevant effect? That’s too ambitious. Therefore, saying the there is no OS difference between CRM>1mm and CRM<1mm (29.8 months vs 18.6 months) is not convincing. It’s true the CI overlap but this may be related to underpower due to small sample size. I would say if this were a real difference, an improvement of survival by about 11 months in pancreas cancer is a big deal.

7. How do the authors explain that median survival of <1mm is better than >1mm and <2mm in table 2 and 3. What is the n for each group?

8. How do the author explain that the recurrence for patients without margin involvement is 41% higher than 15% for positive posterior margin

9. The authors are very conclusive about their findings which is not valid based on a small study prone to false negative findings. Especially when they conclude that there is no OS difference between CRM>1mm and CRM<1mm (29.8 months vs 18.6 months). 11 months difference is clinically relevant difference even if CI overlap.

Reviewer #2: Dear authors,

first of all congratulations to a nice work. I only have minor additional questions.

My questions to you are the following:

1) Do you have preoperative Tumormarker-levels (especially CA19-9) and did you try to include it into your work? Especially for the multiple logistic regression analysis on risk of cancer recurrence it might be another interesting factor.

2) How many patients did recieve adjuvant chemotherapy with nab-Paclitaxel and Gemcitabin and was it perfomed within a clinical trial? To my knowledge the indication is in unresectabel or metastatic pancreatic cancer as it had no overall survival benefit to Gemcitabin-monotherapy in the adjuvant setting (DOI: 10.1200/JCO.2019.37.15_suppl.4000 Journal of Clinical Oncology 37, no. 15_suppl (May 20, 2019) 4000-4000).

3) What imaging technique did you use to evaluate resectability?

4) Based on your findings, would you recommed MRI imaging for the tumours who are likely to involve the medial resection margin for a better preoperative assessment?

Thank you for your answers. I am looking forward to see a multicentric clinical trial to confirm the findings on impact of different resection margins distance or clearance on recurrence and survival.

6. PLOS authors have the option to publish the peer review history of their article (what does this mean?). If published, this will include your full peer review and any attached files.

Reviewer #1: No

Reviewer #2: No

---

## [Author Response · Author response to Decision Letter 0]

3 Feb 2023

Dear Editor, dear Reviewers,

Thank you very much for the constructive feedback. We are happy to report that we agreed with most of the comments and changed the manuscript accordingly. In our opinion, the changes made based on your comments have greatly improved and strengthened the manuscript. For the few points where we disagreed, or perhaps implemented the changes slightly differently than intended, please refer to the point-by-point responses in our rebuttal letter. 

Verena Uslar

For all authors

---

## [Editor Report · Decision Letter 1]

5 Feb 2023

The impact of resection margin distance on survival and recurrence in pancreatic ductal adenocarcinoma in a retrospective cohort analysis

PONE-D-22-07704R1

Dear Dr. Uslar

We’re pleased to inform you that your manuscript has been judged scientifically suitable for publication and will be formally accepted for publication once it meets all outstanding technical requirements.

Kind regards,

Fabian Finkelmeier, M.D.

Academic Editor

PLOS ONE

Additional Editor Comments (optional):

Thank you for reviewing the manuscript according to the comments. We can hereby accept the publication in my opionion in the reviewed form.

---

## [Editor Report · Acceptance letter]

8 Feb 2023

PONE-D-22-07704R1 

The impact of resection margin distance on survival and recurrence in pancreatic ductal adenocarcinoma in a retrospective cohort analysis 

Dear Dr. Uslar:

I'm pleased to inform you that your manuscript has been deemed suitable for publication in PLOS ONE. Congratulations! Your manuscript is now with our production department. 

Kind regards, 

on behalf of

PD. Dr. Fabian Finkelmeier 

Academic Editor

PLOS ONE